# Evaluation of a Point-of-Need Molecular Diagnostic Tool Coupled with Rapid DNA Extraction Methods for Visceral Leishmaniasis

**DOI:** 10.3390/diagnostics13243639

**Published:** 2023-12-11

**Authors:** Prakash Ghosh, Rajashree Chowdhury, Khaledul Faisal, Md. Anik Ashfaq Khan, Faria Hossain, Md. Abu Rahat, Md. Arko Ayon Chowdhury, Nishad Tasnim Mithila, Mostafa Kamal, Shomik Maruf, Rupen Nath, Rea Maja Kobialka, Arianna Ceruti, Mary Cameron, Malcolm S. Duthie, Ahmed Abd El Wahed, Dinesh Mondal

**Affiliations:** 1Nutrition Research Division (NRD), International Centre for Diarrhoeal Disease Research, Bangladesh (icddr,b), Dhaka 1212, Bangladesh; chowdhury_rajashree@yahoo.com (R.C.); otivuj@gmail.com (K.F.); faria109@gmail.com (F.H.); abu.rahat17@gmail.com (M.A.R.); arko.bmb.du@gmail.com (M.A.A.C.).; nishad.mithila@northsouth.edu (N.T.M.); mkamalfhdu@gmail.com (M.K.); shomik_stj@yahoo.com (S.M.); rupennath77@gmail.com (R.N.); din63d@icddrb.org (D.M.); 2Institute of Animal Hygiene and Veterinary Public Health, Leipzig University, D-04103 Leipzig, Germany; anik.ashfaq@gmail.com (M.A.A.K.); rea_maja.kobialka@vetmed.uni-leipzig.de (R.M.K.); arianna.ceruti@uni-leipzig.de (A.C.); 3London School of Hygiene and Tropical Medicine, University of London, London WC1E 7HT, UK; mary.cameron@lshtm.ac.uk; 4HDT Bio, Seattle, WA 98102, USA; malcolm.duthie@hdt.bio

**Keywords:** visceral leishmaniasis (VL), point-of-need diagnosis, rapid DNA extraction, recombinase polymerase amplification (RPA), real-time PCR

## Abstract

A rapid, cost-effective, and simple nucleic acid isolation technique coupled with a point-of-need DNA amplification assay is a desirable goal for programmatic use. For diagnosis of Visceral Leishmaniasis (VL), Recombinase Polymerase Amplification (RPA) rapid tests for the detection of Leishmania DNA are versatile and have operational advantages over qPCR. To facilitate the delivery of the RPA test at point-of-need for VL diagnosis, we compared two rapid DNA extraction methods, SwiftDx (SX) and an in-house Boil and Spin (BS) method, coupled with RPA amplification, versus more widely used methods for DNA extraction and amplification, namely Qiagen (Q) kits and qPCR, respectively. A total of 50 confirmed VL patients and 50 controls, matched for age and gender, were recruited from Mymensingh, Bangladesh, a region highly endemic for VL. Blood samples were collected from each participant and DNA was extracted using Q, SX and BS methods. Following DNA extraction, qPCR and RPA assays were performed to detect *L. donovani* in downstream analysis. No significant differences in sensitivity of the RPA assay were observed between DNA extraction methods, 94.00% (95% CI: 83.45–98.75%), 90% (95% CI: 78.19–96.67%), and 88% (95% CI: 75.69–95.47%) when using Q, SX, and BS, respectively. Similarly, using qPCR, no significant differences in sensitivity were obtained when using Q or SX for DNA extraction, 94.00% (95% CI: 83.45–98.75%) and 92.00% (80.77–97.78%), respectively. It is encouraging that RPA and qPCR showed excellent agreement (k: 0.919–0.980) when different extraction methods were used and that the DNA impurities using BS had no inhibitory effect on the RPA assay. Furthermore, significantly higher DNA yields were obtained using SX and BS versus Q; however, a significantly higher parasite load was detected using qPCR when DNA was extracted using Q versus SX. Considering the cost, execution time, feasibility, and performance of RPA assay, rapid extraction methods such as the Boil and Spin technique appear to have the potential for implementation in resource-limited endemic settings. Further clinical research is warranted prior to broader application.

## 1. Introduction

Visceral Leishmaniasis (VL), colloquially known as kala-azar, is caused by an intracellular protozoan parasite *Leishmania donovani* that is transmitted to mammalian host by infected female sandflies [1]. Among all neglected tropical diseases, VL is ranked second in mortality and fourth in morbidity, with an annual VL incidence of 50,000–90,000 with around 30% of the cases reported from Bangladesh, India, and Nepal [2,3]. Persisting fever, splenomegaly, weight loss, fatigue, hypergammaglobulinemia, and pancytopenia are the common clinical complications associated with VL, and case fatality is absolute if the patients are left untreated [1]. Since 2005, Bangladesh, India, and Nepal jointly launched the Kala-azar elimination program (KEP), intended to eliminate this deadly infectious disease from the Indian subcontinent by 2015, which was further extended to 2020 [4]. With sustained efforts, Bangladesh has already achieved the elimination goal set by the KEP and the national kala-azar elimination programme is waiting for the verification of elimination by the WHO. Despite the continued success of the KEP, VL experts suggest that a highly sensitive and less-resource-demanding diagnostic tool is imperative to accurately assess the residual infection rates during the peri/post-elimination era to avert further transmission and ensure the sustained success of the program [5,6].

Diagnosis of VL still relies mostly on clinical manifestations along with parasitological or immunological procedures [7,8]. Direct parasite visualization through microscopy of spleen, bone marrow, or lymph node aspirates has been considered the gold standard for VL diagnosis; however, the performance of these direct methods is dependent on the sampling procedure as well as technical expertise [8,9]. The collection of the sample for direct methods involves invasive procedures with an associated risk of fatality. A number of serological techniques such as DAT, IFAT, and ELISA have been developed to overcome the limitations of microscopy methods [7,10] but these methods are incapable of differentiating between past and new infections [11]. Further, antigen detection tests like urinary antigen ELISA showed promising diagnostic accuracy, however, large-scale clinical studies are still required prior to broader application [12]. Additionally, genosensor/biosensor-based methods are still in the development phase and need to go through clinical validation phases to overcome present limitations [13].

With the advancement of molecular technologies, several PCR-based approaches for detecting *L. donovani* DNA in samples from VL patients have been established and are widely used for diagnosis, treatment monitoring, and cure assessment [14]. Despite the promising diagnostic performance of these methods, due to the need for well-equipped facilities and qualified personnel, their application in resource-constrained settings is limited.

To address the shortcomings of the existing molecular methods, and bring the molecular method from the bench to bedside, numerous isothermal amplification methods including Loop-Mediated Isothermal Amplification (LAMP) and Recombinase Polymerase Amplification (RPA) have recently been developed, which offer multifarious advantages over conventional PCR and real-time PCR [15,16,17]. Among the isothermal amplification methods, RPA has gained popularity as it amplifies nucleic acid at a constant temperature and provides results within 15 min. Moreover, the RPA method requires less expensive reagents and minimum laboratory setup with simple equipment. Discerning the diverse advantages of RPA, we evaluated a newly developed RPA assay for the detection of *L. donovani* infection and reported absolute sensitivity and specificity compared to real-time PCR for diagnosis of VL and PKDL [16]. To make this RPA method feasible for point-of-need deployment, we incorporated the assay in a suitcase laboratory and successfully implemented the assay in primary healthcare settings for real-time diagnosis of suspected VL patients [9,18]. With the auspices of a multi-country diagnostic study, this point-of-need system has been established in VL-endemic countries including India, Nepal, and Sri Lanka [18].

To make this isothermal assay even more applicable for field use, and reduce the cost for broader applications, a simple DNA extraction method is required. A spin-column-based extraction method (Qiagen) is widely used but it is very costly and requires a long incubation time and a well-equipped laboratory. To address these constraints, efforts are ongoing to develop a rapid, field-feasible, and cost-effective DNA extraction method [19,20]. In our previous study, we showed that a magnetic-bead-based extraction method, SwiftDx, isolates nucleic acid from clinical samples within 20 min without requiring a high-speed centrifuge. Furthermore, the rapid extraction method provided equivalent clinical sensitivity and specificity to the standard spin-column-based extraction method when using both real-time PCR and RPA assay [16]. In addition, we have developed an in-house Boil and Spin (BS) DNA extraction method, which requires minimum instruments and equipment and uses less expensive reagents. In earlier studies, these DNA isolation methods coupled with qPCR and isothermal assays were investigated and shown to have promising diagnostic performance for detecting PKDL and CL cases [18,21,22]. Therefore, in the quest to develop an accurate, rapid, and field-feasible diagnostic tool, we further investigated the performance of these rapid DNA extraction methods, coupled with RPA and qPCR amplification techniques, for detecting *L. donovani* DNA in patient blood to diagnose VL cases in endemic areas in Bangladesh.

## 2. Materials and Methods

### 2.1. Study Sites and Populations

Field and laboratory activities were conducted. Field activities were performed at Surya Kanta Kala-azar Research Centre (SKKRC), Mymensingh, Bangladesh, a region highly endemic for VL, and laboratory activities were conducted in the Emerging Infections and Parasitology Laboratory, icddr,b, Dhaka, following the approval of icddr,b IRB. Fifty treatment-seeking suspected VL cases residing in the endemic region were enrolled at SKKRC, the only specialized hospital for treatment of VL and related complications. All patients were diagnosed based on the national guidelines for VL by a hospital physician. According to the guidelines, any individual from an endemic area with fever for more than two weeks, splenomegaly, and a positive rk39 test will be considered as a VL patient. Further, all the patients enrolled in this study were also positive in the direct agglutination test (DAT). Following the initial examination, each patient was invited to participate in the study and written informed consent was obtained from either the participant or the legal guardian of child participants before samples were collected. Following standard procedures, the study physician collected blood samples from each participant. All VL patients were referred for treatment according to national guidelines and each was found to be responsive to treatment.

### 2.2. Study Design

Three methods of DNA extraction, Spin column (Qiagen, Q), SwiftDX (SX), and Boil and Spin (BS), were used to extract DNA from whole blood samples of 50 confirmed VL cases and 50 healthy volunteers from the same endemic region (Figure 1). Following DNA extraction using Q and SX methods, *L. donovani* DNA was detected in the same blood samples using both qPCR and RPA amplification methods but only RPA amplification was used following DNA extraction using the BS method. Thus, five treatment pairings (Q-qPCR, Q-RPA, SX-qPCR, SX-RPA and BS-RPA) were available to compare their sensitivity and specificity using the methods below.

### 2.3. Sample Collection and Storage

Blood specimens were collected through venipuncture by a trained phlebotomist at SKKRC following proper precautions. Three mL of blood was placed in an EDTA-containing vacutainer and then transported to icddr,b under cold chain conditions. All laboratory tests were performed at icddr,b.

### 2.4. DNA Extraction from Clinical Specimens following Three DNA Extraction Methods

#### 2.4.1. Qiagen DNA Extraction Method

DNA was isolated from 200 μL whole blood sample using a QIAamp DNA tissue and blood mini kit (Qiagen, Hilden, Germany) according to the manufacturer’s protocol. Briefly, 200 μL whole blood was added to 20 μL protease K and then 200 μL buffer AL was also added. After vortexing, the mixture was incubated at 56 °C for 10 min. A total of 200 μL 100% ethanol was added to the mixture and transferred to the QIAamp mini spin column. Following washing steps, the DNA product was eluted with elution buffer to acquire the final DNA solution.

#### 2.4.2. SwiftDx DNA Extraction Method

A rapid and simple blood lysis protocol (Xpedite, Munich, Germany), SwiftDx—established previously—was followed as described: A total of 500 μL of whole blood was incubated with 1500 μL of the enrichment buffer and 30 μL of the magnetic beads for three minutes at room temperature [16]. Then, the magnetic beads were separated using a magnetic stand and the supernatant was removed without disturbing the beads. After that, the beads were washed twice with 500 μL enrichment buffer. Thereafter, 100 μL of the lysis buffer was added and the mixture was incubated at 95 °C for 10 min. The beads were then separated using magnetic beads and the supernatant was collected for further assays.

#### 2.4.3. Boil and Spin DNA Extraction Method

A previously published extraction procedure was slightly modified as follows: A total of 60 μL of whole blood (heparin treated) was transferred to an extraction tube containing 60 μL of extraction buffer (400 mM NaCl, 40 mM Tris pH 6.5, 0.4% SDS), which was mixed by vortexing for 10 s [23]. Then, the whole blood sample, along with the extraction buffer, was incubated in a heat block at 95 °C for 5 min. After the incubation period, the mixture was centrifuged for 3 min at 10,000× *g* and 30 μL of clear supernatant was transferred to the Dilution Tube containing 345 μL of PCR-grade water. After extraction, DNA samples were stored at −20 °C for the following assays.

### 2.5. DNA Purity and Concentration

Using a Thermo Scientific Nanodrop™ 2000 Spectrophotometer (Thermo Scientific, Bremen, Germany), DNA concentration/quantity was determined from its OD value at 260 nm. The purity of each extracted DNA sample was assessed through the ratio of OD value at 260 nm and 280 nm where the standard ratio for purified DNA ranges between 1.8 and 2.0.

### 2.6. Molecular Detection of Leishmania Donovani DNA

#### 2.6.1. Recombinase Polymerase Amplification (RPA) Assay

The RPA assay was performed using DNA template extracted from the same blood samples by all three extraction methods using a previously published protocol [22]. In summary, 50 μL of total reaction volume was prepared to perform the assay using a TwistAmp exo kit (TwistAmp exo kits, TwistDx, Cambridge, UK). Master mix was prepared with 420 nM of RPA primer, 120 nM of RPA Probe, 1× rehydration buffer in a tube per sample, and was transferred to the RPA lyophilized pellet. Then, 14 mM Magnesium acetate was pipetted into each tube lid. Following, template DNA was added to the tubes and the tubes were mixed properly after closing the lids. The tubes were immediately placed into the tube scanner (Twista, TwistDx, Cambridge, UK) and incubated for 15 min at 42 °C. The emitted fluorescence signals were measured at 20 s intervals. A combined threshold and first derivative analysis was used for signal interpretation. From master mix preparation to the completion of the assay, the total reaction time for the RPA assay was approximately 20 min.

#### 2.6.2. Real-Time PCR (qPCR)

Targeting a conserved region of Leishmania REPL repeats (L42486.1) specific for *L. donovani* and *L. infantum*, Taqman primers and probes were designed to perform the real-time PCR following a method described in previous studies [20]. Briefly, 20 μL reaction mix was prepared containing 5 μL template, 10 μL of TaqMan^®^ Gene Expression Master Mix (2×), 1 μL pre-ordered primer-probe mix, and PCR-grade water. Amplification was performed on a Biorad CFX96 icycler system with the following reaction conditions: A total of 10 min at 95 °C, followed by 45 cycles of 15 s at 95 °C, and 1 min at 60 °C. The total reaction time for real-time PCR was approximately 120 min. Specimens ran as duplicates. To quantify the parasite load, each run included one standard curve with DNA concentration corresponding to parasite load of 10,000 to 0.1 parasites per reaction. Each run also included one reaction with molecular-grade water as a negative control. Samples with cycle threshold (Ct) > 40 were considered negative.

### 2.7. Statistical Analysis

All statistical analyses were performed using GraphPad Prism (Version 8.1.2), SPSS (Version 20.0), and online MedCalc statistical software (https://www.medcalc.org/calc/diagnostic_test.php). Based on distribution of data, both parametric and non-parametric tests were performed. Absolute numbers and percentages were used to represent categorical variables. Clinical sensitivity and specificity were determined with 95%CI according to the standard method for statistical analyses. Cohen’s Kappa and McNemar’s tests were performed to find out the concordance and discordance among three DNA extraction methods when combined with RPA/qPCR assay. Further, Mann–Whitney’s U test was performed to measure the differences between the DNA yields of the three extraction methods. Pearson’s correlation coefficient was used to determine the relationship of Ct values and parasite burden between Q-qPCR and SX-qPCR assays. A *p*-value < 0.05 was considered statistically significant.

## 3. Results

### 3.1. Descriptive Characteristics of Study Participants

All of the 50 suspected VL cases who participated in this study following clinical examination presented with fever for more than two weeks with splenomegaly. All of the cases were found to be positive using rK39 RDT and DAT tests. The majority of participants were male (56.0%) and the median age was 35.5 (IQR: 21.50–50.00) years. Almost one-third of the patients reported previous history of VL and hepatomegaly (Table 1). Interestingly, more than half of the suspected VL cases exhibited pancytopenia (Table 1). All of the 50 endemic controls involved in this study were healthy (Table 1).

### 3.2. Diagnostic Performance of Molecular Assays with Different Extraction Methods

In the case of the qPCR assay, superior sensitivity was achieved using DNA extracted through both Qiagen and SwiftDx methods, which was 94.00% (95% CI: 83.45–98.75%) and 92.00% (80.77–97.78%), respectively. For the RPA assay, a sensitivity of 92.00% (95% CI: 80.77–97.78%) and 90.00% (95% CI: 78.19–96.67%) was found for DNA isolated with the Qiagen and SwiftDx methods, respectively (Table 2). When the Boil and Spin DNA extraction method was followed, the RPA assay also presented an elevated sensitivity of 88.00% (75.69–95.47%). Each of the molecular assays accomplished absolute specificity with each of the DNA extraction methods.

All three DNA extraction methods showed excellent agreement (Table 3) when parasite DNA was detected using both qPCR and RPA assays.

Out of the 50 confirmed VL cases 44 cases were found to be positive by all of the methods (Figure 2). Statistically significant differences were not observed among the diagnostic performance of qPCR and RPA assays with three different DNA extraction methods.

### 3.3. Effect of DNA Yields on Enumeration of Parasitemia through Molecular Methods

Significant differences (*p* < 0.0001) were found in the mean concentration of DNA between three different DNA extraction procedures. The highest DNA yield was obtained using SX, with a mean DNA concentration of 170.4 ng/µL, whereas for BS and Q the mean DNA concentrations were 26.91 ng/µL and 13.10 ng/µL, respectively. In terms of DNA purity, a higher mean 260/280 absorbance ratio of 1.82 was obtained using Q, whereas a comparatively poor mean 260/280 absorbance ratio of 0.59 was presented by the SX method (Table 2). Moreover, significant discordance (*p* < 0.0001) was observed among the concentration of DNA isolated through the three different extraction methods (Figure 3A).

Based on the parasite load detected using qPCR, there was a significant difference (*p* < 0.0001) between DNA extracted through the Qiagen and SwiftDx methods. A greater parasite burden was measured in 84% (42/47) of qPCR-positive cases with DNA extracted using Qiagen, whereas only five cases exhibited an increased parasite load when DNA was isolated using the SX method. The mean parasite loads for Q-qPCR and SX-qPCR were 669.4 and 217.8 parasites/mL, respectively (Figure 3B). While a strong positive correlation (r = 0.763, *p* < 0.0001) was found in Ct values between Q-qPCR and SX-qPCR, a weak correlation (r = 0.343, *p* < 0.01) was observed in the parasite load obtained using Q-qPCR and SX-qPCR assays. On the other hand, however, no significant difference was found in TT values among Q-RPA, BS-RPA, and SX-RPA.

### 3.4. Comparative Analysis of Different DNA Extraction Methods

The key features of each DNA isolation method are summarized in Table 4. The time to obtain a result includes the duration for sample processing, while the kit cost comprises exclusively the reagent cost for sample processing, which includes extraction.

## 4. Discussion

Despite the unparalleled success of the Kala-azar elimination programme in Bangladesh, ensuring the zero-transmission goal and preventing possible resurgence of VL remains an endgame challenge. VL experts and the Diagnostics Technical Advisory Group (DTAG) suggested the development of a highly sensitive, cost-effective, and field-feasible diagnostic for early detection of residual LD infection. With the advancement of the Nucleic Acid Amplification Tests (NAATs)-based methods, numerous isothermal amplification methods such as LAMP and RPA have also been developed [26]. Notwithstanding the promising diagnostic efficiency of isothermal techniques, most are yet to be standardized for large-scale implementation in endemic settings. Moreover, the lack of a rapid, cost-effective, and simple DNA isolation technique precludes the field deployment of the isothermal assays for the detection of Leishmaniasis. Therefore, considering the unmet needs, in our previous studies we evaluated the efficiency of a rapid DNA extraction method (SwiftDx) coupled with a Recombinase polymerase amplification (RPA) assay for detecting LD parasites with a limited number of buffy coat and skin biopsy samples [16,22]. The findings of the former studies inspired us to apply the rapid extraction methods towards facilitating the diagnosis of VL in remote settings. Since venous blood offers a suitable specimen for detecting circulating parasites in VL patients, we undertook the present study with an overarching goal of evaluating the efficiency of a magnetic-bead-based method and another in-house Boil and Spin rapid DNA extraction method combined with qPCR and RPA assays.

In the present study, when the magnetic-bead-based (SwiftDx) and the in-house Boil and Spin extraction methods were performed, the RPA assay generated promising sensitivities of 90.0% (95%CI:78.19–96.67%) and 88.0% (95% CI: 75.69–95.47%), respectively. A study conducted by Mondal et al. showed an absolute sensitivity for the RPA assay with the SwiftDx DNA extraction method, however, the sample size of the former study was limited [16]. In the current study, for the first time, an in-house Boil and Spin DNA extraction method coupled with RPA assay has been evaluated to detect LD parasites in blood samples. To date, a handful of studies have reported the promising performance of the in-house Boil and Spin Extraction method combined with the LAMP assay (Table 4). The elevated sensitivity of the in-house extraction method might be attributed to the multiple target amplicons (kDNA and 18SrDNA) that are used in the LAMP assay. In parallel to the findings of our previous study, when the standard spin-column-based extraction method was performed, the RPA assay showed the highest sensitivity of 92.0% (95%CI: 80.77–97.78%) (Table 4) [18].

As anticipated, the qPCR assay presented a superior sensitivity of 94.0% (95%CI: 83.45–98.75%), with the DNA extracted through the reference spin-column-based method, which is consistent with the findings of our previous study [18]. Here, for the first time, we evaluated the performance of a rapid SwiftDx DNA extraction method coupled with the qPCR method, which exhibited an elevated sensitivity of 92.0% (95%CI: 80.77–97.78%).

Out of 50 clinically confirmed VL cases, 47(94%) patients were positive for *L. donovani* DNA using qPCR and/or RPA assays in combination with the three different DNA extraction methods from whole blood samples. Among the positive cases, 46 patients were detected using both qPCR and RPA assays, where only one VL case was found positive exclusively in the qPCR assay. Similar to previous studies, we observed excellent agreement between qPCR and RPA assays in the detection of leishmania DNA (Table 3). In contrast, 44 cases were detected through the RPA assay coupled with reference and two rapid DNA extraction methods (Figure 2). However, in sporadic cases, the RPA assay has been found to fail in detecting leishmania DNA in clinical samples with low parasitemia or high Ct values in the qPCR. Without exception, in the current study, the RPA assay could not detect the VL cases with low parasite burden. Such a caveat in the RPA reaction can be attributed to the viscous nature of the crowding agents that impedes the diffusion of reagents through the reaction mixture and inherently increases the amplification time, which renders a negative impact on the RPA performance for the clinical samples having a low number of DNA templates or parasites [27].

As expected, the Q-qPCR assay quantified more parasites than that of the rapid extraction method (Figure 3B). The higher efficiency of the qPCR assay for the spin-column-based extraction method can be attributed to the more purified DNA being obtained through this reference extraction method, whereas the impurities in DNA followed by rapid DNA extraction methods inhibit the amplification in qPCR assays. Despite this, we observed an equivalent efficiency of the RPA assay for both reference and rapid extraction methods (Table 3). The RPA reaction is comparatively robust as the impurities—including haemoglobin, protein, or phenolic contaminants—have no interference with the enzymes being involved in the amplification of the target DNA. Moreover, the RPA assay is less resource-demanding, simple, and less time-consuming, and the reagents used in the assay are comparatively cheap. In our current study, we estimated the cost for rapid extraction methods as USD 0.006 and USD 1.70 per sample for the Boil and Spin and SwiftDx, respectively, whereas the reference spin-column-based method was found to be costly (USD 3.4) as it requires an expensive commercial kit (Table 5). Among the rapid extraction methods, the Boil and Spin method requires minimum resources, involves less execution time, and the method is performed with in-house buffers, which do not require any commercial kit to isolate the DNA from the clinical samples.

Regardless of the promising findings, this study has a few limitations that could and should be addressed in future studies. Firstly, the sample size was relatively small, and this may impact the diagnostic performance of the investigative methods. However, in the wake of this study, for the first time, we performed a comparative analysis of qPCR and RPA assays combined with three DNA extraction methods. It is also worth noting that in earlier studies, the RPA assay combined with rapid extraction methods were evaluated for the diagnosis of cutaneous and post-kala-azar dermal Leishmaniasis, whereas in the current study, the sample matrix differed from that of previous studies [21,22]. The second drawback is that the qPCR assay was not performed with DNA extracted through the Boil and Spin method. Reports by Chowdhury et al. and Hossain et al. stated that the Boil and Spin method is not compatible with qPCR due to the presence of impurities (e.g., hemoglobulin, protein), which causes PCR reaction inhibition [22,25]. Finally, all of the healthy controls in this study were enrolled mainly based on RDT and DAT results, which might overestimate the diagnostic performance of the assays. However, the absolute specificity of the assays can be attributed to the sustained low prevalence of the disease due to the holistic efforts of the Kala-azar elimination programme in Bangladesh. Moreover, in terms of specificity, the findings of earlier studies are in concordance with the present study [16,18,22].

Following the London declaration, efforts are ongoing to eliminate Visceral Leishmaniasis and other neglected tropical diseases. The road map for NTDs 2021–2030 has further strengthened the control measures to eliminate this vicious disease, which predominately affects poor communities in both new and old worlds [28]. Despite significant improvement in disease elimination, an ultra-sensitive method is still required to gauge residual infection at the peri- and post-elimination periods in the Indian subcontinent. Furthermore, a handful of African countries are still impacted with a high burden of the disease, and since the RPA assay is less resource-demanding and highly sensitive, the recently developed pan-leishmania assay could eventually be implemented in those endemic regions [29]. Of further benefit, a number of RPA assays have recently been combined into a lateral flow format that is implementable in resource settings [30,31,32,33]. However, there is still a need for cost-effective, field-feasible, and rapid DNA extraction methods for downstream analysis. Therefore, incorporation of the newly devised DNA extraction methods would facilitate the detection of Leishmaniasis in different resource-constrained settings. Here to note, the RPA assay combined with the rapid DNA extraction method has surpassed the qualification of a diagnostic test (sensitivity + specificity ≥ 1.5 value) set by Power et al. [34].

Considering our findings, we recommend that the RPA assay coupled with SwiftDx and the Boil and Spin DNA extraction method serve as an alternative diagnostic method for routine diagnosis of Visceral Leishmaniasis in low-resource settings. Considering the time, cost, and operational complexity, the in-house Boil and Spin extraction method coupled with the RPA assay should be prioritized for point-of-need detection of VL cases in resource-constrained endemic settings. It is encouraging that the rapid extraction method coupled with the RPA assay meets most of the criteria set by the WHO for diagnostics in resource-limited settings (ASSURED), thereby ensuring affordability, sensitivity, specificity, user-friendliness, rapidity, robustness, equipment-free methods, and ensuring the feasibility of delivering the test to end users. Considering the multifarious advantages, this diagnostic algorithm can be applied as a surveillance tool during the post-elimination period for gauging the residual infection in the endemic hotspots. For surveillance, VL cases and their household members might be under investigation to track the ongoing transmission. However, the approach of surveillance should be adopted by the country-specific VL elimination programs. Nevertheless, taking the propitious study findings into account and the prior application of the rapid DNA extraction methods, proper optimization is a prerequisite. Lastly, a future large-scale multi-country study should be performed to establish these rapid extraction methods coupled with the RPA assay as a comprehensive field-based point-of-need diagnostic tool for the detection of VL.

## Figures and Tables

**Figure 1 diagnostics-13-03639-f001:**
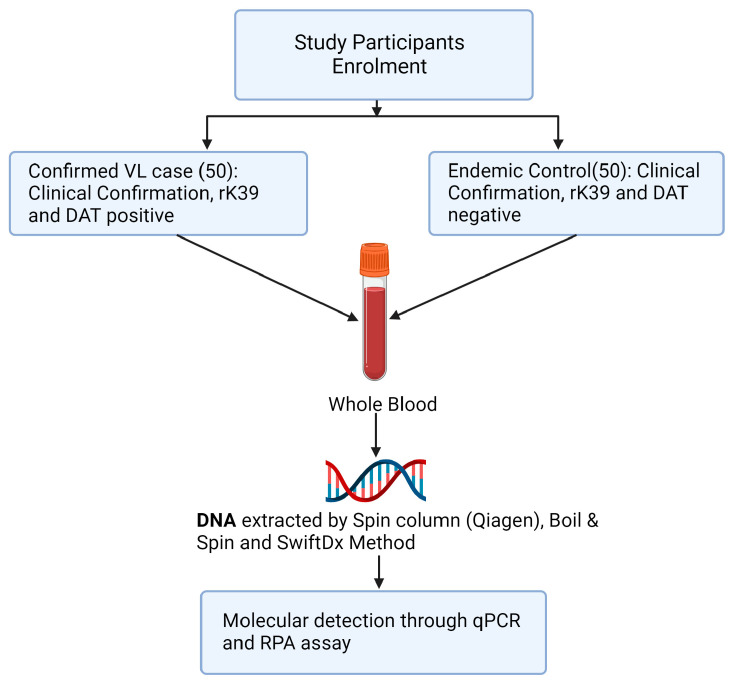
Study design flowchart.

**Figure 2 diagnostics-13-03639-f002:**
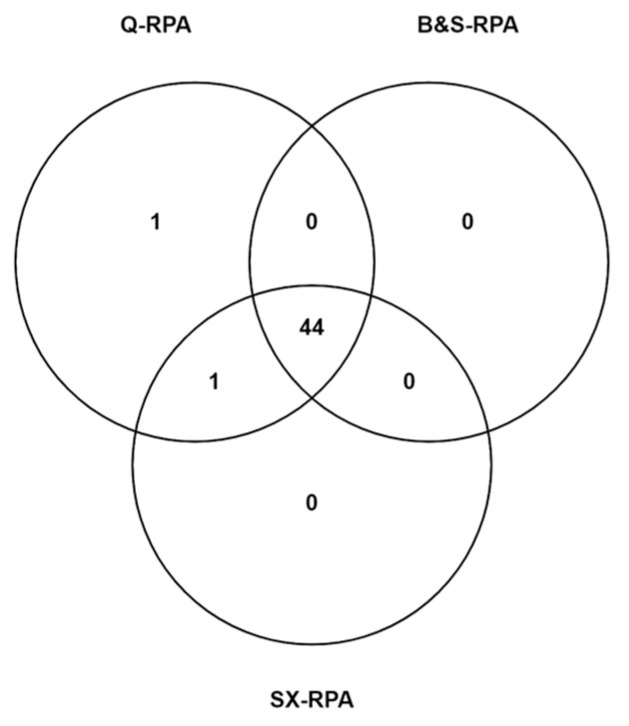
Venn diagram represents the distribution of 46 VL cases that were detected through different DNA extraction methods in combination with RPA assay. Among 46 VL cases, 44 cases were positive using RPA assay coupled with Qiagen, Boil and Spin, and SwiftDx DNA extraction methods, whereas only 1 case was exclusively detected through Q-RPA assay.

**Figure 3 diagnostics-13-03639-f003:**
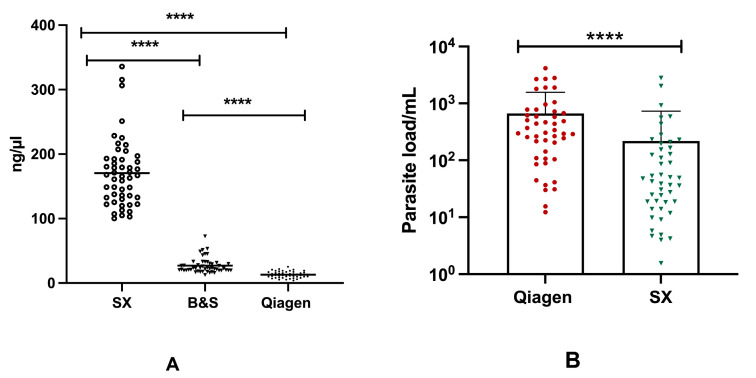
Concentration of DNA in different extraction assays and differences in the parasite load in qPCR assay: (**A**) Concentration of DNA isolated through Spin-column (Qiagen), Boil and Spin (BS) and SwiftDx (SX) methods. (**B**) Distribution of parasite burden between Q-qPCR and SX-qPCR. [* represents the significance level (**** corresponds to the *p* value of <0.0001)].

**Table 1 diagnostics-13-03639-t001:** Clinical and Demographic parameters of the study participants (*N* = 100).

Variable	Case (*n* = 50)	Control (*n* = 50)
Male, *n*/*N* (%)	28/50 (56.0%)	29/50 (58.0%)
Age in years, mean ± SD	31.50 ± 14.99	27.64 ± 14.21
Past history of VL, *n* (%)	15/50 (30.0%)	0 (0%)
Fever more than two weeks, *n* (%)	50/50 (100.0%)	0 (0%)
Splenomegaly, *n* (%)	50/50 (100.0%)	0 (0%)
Hepatomegaly, *n* (%)	14 (28.0%)	0 (0%)
Pancytopenia, *n* (%)	28 (56.0%)	0 (0%)
rk39 RDT positive, *n* (%)	50/50 (100.0%)	0 (0%)
DAT positive, *n* (%)	50/50 (100.0%)	0 (0%)

**Table 2 diagnostics-13-03639-t002:** Analysis of performance for three DNA extraction methods with qPCR and RPA assays.

DNA Extraction Methods	Mean OD 260/280 Ratio ± SD [95% CI] *N* = 50	Mean DNA Conc. (ng/µL) ± SD [95% CI] *N* = 50	Sensitivity of qPCR [95% CI] (*n*/*N*) *N* = 50	Sensitivity of RPA [95% CI] (*n*/*N*) *N* = 50	Specificity of qPCR and RPA [95% CI] (*n*/*N*) *N* = 50
Qiagen	1.82 ± 0.22 [1.76–1.88]	13.10 ± 4.72 [11.75–14.45]	94.00% [83.45–98.75%] (47/50)	92.00% [80.77–97.78%] (46/50)	100.00% [92.89–100.00%] (0/50)
SwiftDx (SX)	0.59 ± 0.11 [0.57–0.63]	170.4 ± 52.69 [155.4–185.4]	92.00% [80.77–97.78%] (46/50)	90.00% [78.19–96.67%] (45/50)	
Boil and Spin (BS)	2.02 ± 0.49 [1.88–2.22]	26.91 ± 11.69 [23.58–30.23]		88.00% [75.69–95.47%] (44/50)	

**Table 3 diagnostics-13-03639-t003:** Concordance and discordance among three DNA extraction methods coupled with qPCR and RPA assays.

Assays	Kappa (k)	Agreement	McNemar (*p*-Value)
Q-qPCR vs. SX-qPCR	0.980	Excellent	0.50
Q-qPCR vs. Q-RPA	0.980	Excellent	1.00
Q-qPCR vs. BS-RPA	0.940	Excellent	0.25
Q-qPCR vs. SX-RPA	0.960	Excellent	0.50
SX-qPCR vs. Q-RPA	0.960	Excellent	1.00
SX-qPCR vs. BS-RPA	0.919	Excellent	1.00
SX-qPCR vs. SX-RPA	0.940	Excellent	1.000
Q-RPA vs. BS-RPA	0.960	Excellent	0.500
Q-RPA vs. SX-RPA	0.980	Excellent	1.000
SX-RPA vs. BS-RPA	0.980	Excellent	1.000

**Table 4 diagnostics-13-03639-t004:** Comparing sensitivity and specificity of present study with earlier studies using different DNA extraction and amplification methods.

References	Country	Study Population	Reference Test	Clinical Specimen Tested	DNA Isolation Assay	Molecular Technique	Sensitivity	Specificity
Mondal et al. (2016) [16]	Bangladesh	23 VL, 20 PKDL cases, and 5 Asymptomatic	qPCR	Buffy Coat, Skin Biopsy	Qiagen DNeasy Blood and tissue kit, SwiftDx (SX)	RPA	100%	100%
Gunaratna et al. (2018) [21]	Sri Lanka	150 suspected CL cases	Microscopy, Conventional PCR	Direct Skin Punch Biopsy	SwiftDx (SX)	RPA	65.5%	100%
				Skin Punch Biopsy-ATL	Qiagen DNeasy Blood and tissue kit	Conventional PCR	92.4%	
				Skin Punch Biopsy-RNAlater			63.4%	
Mukhtar et al. (2018) [24]	Sudan	198 VL suspected cases	Microscopy of lymph node aspirates	Whole Blood	QIAamp mini kit (QIAGEN)	LAMP	100%	99.01%
				Buffy Coat			97.62%	
				Whole Blood	Boil and Spin (BS)		97.62%	
				Buffy Coat			95.24%	
L. S. Batalini et al. (2020) [19]	Brazil	30 Volunteers with negative serology for Leishmaniasis	Serology	Peripheral Blood	20% sodium dodecyl sulfate (SDS), Guanidine isothiocyanate-phenol-chloroform (GTPC), and Commercial kit (GE Healthcare GenomicPrep Blood DNA Isolation Kit^TM^)	Conventional PCR	100%	100%
Chowdhury R et al. (2020) [22]	Bangladesh	30 PKDL cases and 30 endemic controls	Clinical evaluation and rK39 rapid diagnostic	Skin punch biopsy	Spin-column-based method (Qiagen)	qPCR	86.67%	100%
					Spin-column-based method (Qiagen)	RPA	93.33%	
					Boil and Spin (BS)		76.67%	
					SwiftDx (SX)		63.33%	
Hossain et al. (2021) [25]	Bangladesh	80 cases and 80 controls	Clinical evaluation and Rk39 rapid diagnostic test	Whole blood	QIAamp mini kit (Qiagen)	qPCR	72.5%	100%
					Boil and Spin (BS)		-	
					QIAamp mini kit (Qiagen)	LAMP	85%	100%
					Boil and Spin (BS)		96.2%	
Present Study	Bangladesh	50 cases and 50 controls	Clinical evaluation, rK39 rapid diagnostic test and DAT	Whole blood	Spin-column method (Qiagen)	qPCR	94%	100%
					SwiftDx (SX)		92%	
					Spin-column method (Qiagen)	RPA	92%	
					SwiftDx (SX)		90%	
					Boil and Spin (BS)		88%	

**Table 5 diagnostics-13-03639-t005:** Comparison between different DNA extraction methods.

Aspect	Qiagen	Boil and Spin	SwiftDx
Principle	Silica membrane-based spin column	Boiling and centrifugation	Magnetic-bead-based extraction
Volume of sample required	200 μL whole blood	60 μL whole blood	500 μL whole blood
Workflow simplicity	Multiple steps	Fewer steps	Fewer steps
* Turnaround time	50 min	12 min	15 min
* Hands-on time	20 min	4 min	5 min
Downstream applications	Suitable for various applications	Limited applications	Suitable for various applications
** Cost	Relatively high (USD 3.4)	Relatively low (USD 0.006)	Moderate (USD 1.7)

* Inclusive of the time for sample processing. ** Only reagent cost for sample processing.

## Data Availability

The data presented in this study are available on request from the corresponding author. The raw data of the experiments are not publicly available due to the data protection policy of icddr,b.

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
