# Peer review of "Evaluation of a Point-of-Need Molecular Diagnostic Tool Coupled with Rapid DNA Extraction Methods for Visceral Leishmaniasis"

_diagnostics, 2023, doi:10.3390/diagnostics13243639_

Round 1
Reviewer 1 Report
Comments and Suggestions for Authors
Evaluation of a point-of-need molecular diagnostic tool coupled with rapid DNA extraction methods for Visceral Leishmaniasis
The authors compared two different novel rapid methods of DNA extraction (SwiftDx (SX) and an in-house Boil & Spin (B&S) method) combined with a ‘Recombinase Polymerase Amplification’ (RPA) rapid test for detection of Leishmania DNA to classic extraction methods (Qiagen, (Q)) combined with qPCR. When using the RPA, on significant differences were apparent between the three extraction methods, Q, SX and B&S, the same applied when qPCR was used instead of RPA (sensitivity 88% to 94%). Agreement between RPA and qPCR was excellent (kappa > 0.9). DNA yields with SX and B&S were significantly higher than with Q. Rapid extraction methods in combination with RPA offer a promising approach for resource limited settings.
Major comments:
This is a well written and very relevant article. I do have a few issues that I would like the authors to clarify.
Line 245-253: Why did the authors opt for a ROC analysis? A ROC analysis is useful if there are different cutoffs to consider for deciding on whether a sample is positive or negative but here the cutoff was fixed at (Ct) ≤ 40.
Line 268-269: ‘Significant variation (p<0.0001) was found in the mean concentration of DNA between three different DNA extraction procedures.’ This is not clearly described in the methods it seems. I only found that: ‘The Pearson correlation coefficient was used to determine the relationship of Ct values and parasite burden between Q-qPCR and SX-qPCR assays. P-value <0.05 was considered as statistically significant.’ Have I overlooked it?
Line 437-439: ‘Considering our findings, we recommend that the RPA assay coupled with SwiftDx and Boil & Spin DNA extraction method serve as an alternative diagnostic method for routine diagnosis of Visceral leishmaniasis in low-resource settings.’
The gold standard used in this study was fever > 2 weeks, splenomegaly and a positive RDT. All 50 cases thus diagnosed were confirmed with DAT and 47(94%) were positive on at least one molecular method. It seems the very simple diagnostic algorithm still performs extremely well. Is there a need then for replacing the RDT by a molecular method at point of care? I agree that as incidence further declines the positive predictive value of the current diagnostic algorithm needs to be monitored but that doesn’t need to happen at point of care. Why would it not be enough to collect samples on the spot, initiate treatment based on the current diagnostic algorithm and test the samples with molecular methods in a remote laboratory? In that case, would there still be a need for alternative extraction methods?
Line 442-444: ‘It is encouraging that the rapid extraction method meets the WHO standard for diagnostics in resource-limited settings (ASSURED), thereby ensuring affordability, sensitivity, specificity, user-friendliness, rapidity, robustness, equipment-free, and feasibility of delivering the test to end users.’
Are these methods really equipment free?
Minor comments:
Line 72-73: Further, antigen detection tests like urinary antigen ELISA showed promising sensitivity, however, refinement of the assay is required to improve its specificity [14].
Ref 14 Ejazi et al states that: ‘Sensitivity and specificity of urine-ELISA were 97.94% (95/97) and 100% (75/75) respectively, for VL. Importantly, dipstick test demonstrated 100% sensitivity (97/97) and specificity (75/75) in VL diagnosis.’
Line 186-187: Please check sentence: ‘The RPA assay was performed using DNA extracted from the same blood samples by all three-extraction methods DNA using a previously published protocol [26].’
Line 229-230: ‘The majority of participants were male (56.0%) and the mean age was 31.50 ± 14.99 years.’
Unlikely that age was normally distributed. Please present median and IQR.
Line 300: Table 5 seems more appropriate for the introduction since only the last line is really about results of this study. Even more so because it shows that some of these methods have been evaluated before.
Author Response
Reviewer 1:
- The authors compared two different novel rapid methods of DNA extraction (SwiftDx (SX) and an in-house Boil & Spin (B&S) method) combined with a ‘Recombinase Polymerase Amplification’ (RPA) rapid test for detection of Leishmania DNA to classic extraction methods (Qiagen, (Q)) combined with qPCR. When using the RPA, on significant differences were apparent between the three extraction methods, Q, SX and B&S, the same applied when qPCR was used instead of RPA (sensitivity 88% to 94%). Agreement between RPA and qPCR was excellent (kappa > 0.9). DNA yields with SX and B&S were significantly higher than with Q. Rapid extraction methods in combination with RPA offer a promising approach for resource limited settings.
Response: Thanks for the positive comments.
- Line 245-253: Why did the authors opt for a ROC analysis? A ROC analysis is useful if there are different cutoffs to consider for deciding on whether a sample is positive or negative but here the cutoff was fixed at (Ct) ≤ 40.
Response: Thanks for being inquisitive regarding the application of ROC. In this study we did use ROC analysis has been done to determine the diagnostic accuracy of the methods which have been represented with the value of AUC or area under the curve following ROC analysis.
- Line 268-269: ‘Significant variation (p<0.0001) was found in the mean concentration of DNA between three different DNA extraction procedures.’ This is not clearly described in the methods it seems. I only found that: ‘The Pearson correlation coefficient was used to determine the relationship of Ct values and parasite burden between Q-qPCR and SX-qPCR assays. P-value <0.05 was considered as statistically significant.’ Have I overlooked it?
Response: The differences of the DNA yields between the three extraction methods were measured using Mann-Whitney’s U test. It has been added in the methodological part of the revised manuscript.
- Line 437-439: ‘Considering our findings, we recommend that the RPA assay coupled with SwiftDx and Boil & Spin DNA extraction method serve as an alternative diagnostic method for routine diagnosis of Visceral leishmaniasis in low-resource settings.’
The gold standard used in this study was fever > 2 weeks, splenomegaly and a positive RDT. All 50 cases thus diagnosed were confirmed with DAT and 47(94%) were positive on at least one molecular method. It seems the very simple diagnostic algorithm still performs extremely well. Is there a need then for replacing the RDT by a molecular method at point of care? I agree that as incidence further declines the positive predictive value of the current diagnostic algorithm needs to be monitored but that doesn’t need to happen at point of care. Why would it not be enough to collect samples on the spot, initiate treatment based on the current diagnostic algorithm and test the samples with molecular methods in a remote laboratory? In that case, would there still be a need for alternative extraction methods?
Response: Fortunately, all the cases included in the study were perfectly diagnosed through clinical algorithm along with Rk39 positivity for the treatment. Moreover, we did not have relapse or treatment failure cases in the study. Here to note, due to the limited access to PCR techniques, most of the VL cases were being treated based on clinical diagnosis during the study period, including the VL cases with previous treatment history. Since the number of VL cases with previous history, relapse VL cases and treatment failure are significantly high, the patients are going under the treatment based on molecular diagnosis as the guideline of elimination program for VL. Therefore, an alternative test to qPCR is desired that can be doable at remote healthcare settings. However, to bring the method onsite a rapid DNA extraction method is still necessary which can be done with limited laboratory set-up.
- Line 442-444: ‘It is encouraging that the rapid extraction method meets the WHO standard for diagnostics in resource-limited settings (ASSURED), thereby ensuring affordability, sensitivity, specificity, user-friendliness, rapidity, robustness, equipment-free, and feasibility of delivering the test to end users.’
Are these methods really equipment free?
Response: Thanks for putting a spotlight on the issue. Indeed, we need small equipment to perform the RPA assay coupled with rapid extraction method. However, it showed the promises to overcome the shortcomings of the qPCR and standard QIAGEN based DNA extraction method. We have revised the section to keep a concurrence between the study finding and remarks on ASSURED. Please see the revised section (Line 444-446).
- Line 72-73: Further, antigen detection tests like urinary antigen ELISA showed promising sensitivity, however, refinement of the assay is required to improve its specificity [14].
Ref 14 Ejazi et al states that: ‘Sensitivity and specificity of urine-ELISA were 97.94% (95/97) and 100% (75/75) respectively, for VL. Importantly, dipstick test demonstrated 100% sensitivity (97/97) and specificity (75/75) in VL diagnosis.’
Response: Thanks for your comment. We will revise the sentence accordingly. Albeit, the sensitivity and specificity of the urine based ELISA assays are high, large clinical studies are required prior broader application. Please see the revised section (Line 73-74).
- Line 186-187: Please check sentence: ‘The RPA assay was performed using DNA extracted from the same blood samples by all three-extraction methods DNA using a previously published protocol [26].’
Response: The sentence is checked and found correct.
- Line 229-230: ‘The majority of participants were male (56.0%) and the mean age was 31.50 ± 14.99 years.’
Unlikely that age was normally distributed. Please present median and IQR.
Response: The median and the IQR have been added in place of mean value in the revised manuscript.
- Line 300: Table 5 seems more appropriate for the introduction since only the last line is really about results of this study. Even more so because it shows that some of these methods have been evaluated before.
Response: The table has been presented as an exhibit to compare the current study findings with previous ones.
Reviewer 2 Report
Comments and Suggestions for Authors
The study by Ghosh et al evaluated DNA extraction methods from blood samples used for the diagnosis of visceral leishmaniasis. This is an important study since there is still no consensus on how best to extract DNA from small samples of blood for the diagnosis of visceral leishmaniasis. The authors have strong technical expertise in performing the RPA and qPCR assays and consequently the study was technically sound. Overall, the least expensive and simplest extraction procedure; Boil & Spin performed as well or better than the expensive and more complicated procedures; Qiagen and SwiftDx. This is a relevant observation when considering that these assays are typically performed in less than optimum conditions in LMIC countries. Overall this study provides important information in support of the KEP for the development of a field-feasible assay for detecting L. donovani in the blood.
Minor points to consider during revisions.
The enrolment of 50 VL cases appears to be high considering the effectiveness of the KEP. What was the time frame for the enrolment of 50 VL cases?
Past history of VL, 15/50 is 30%, not 50% as indicated in Table 1. Also, 30% with a past history seems high. Were these relapse. Is this now normal in Bangladesh?
What was the target DNA for RPA amplification?
Regarding Yield where comparisons were made by comparing ng/ul. However, the extraction volumes were different for each approach. Was this difference taken into account when calculating the yields?
Figure 4B indicates “Speed Xtract” Is this the same as SwiftDx (SX). These terms should be consistent throughout.
Line 370; should read “47(94%) patients were positive for L. donovani DNA using qPCR and/or RPA assays
In the discussion, it would be helpful to elaborate how the detection of parasite in the blood will support the KEP. Will the test only be performed on positive diagnosed cases (rK39 +) or can it also be done on potential asymptomatic or family members of VL cases.
Comments on the Quality of English LanguageThe quality of the English is very good. I made some suggestions above.
Author Response
Reviewer 2:
- The study by Ghosh et al evaluated DNA extraction methods from blood samples used for the diagnosis of visceral leishmaniasis. This is an important study since there is still no consensus on how best to extract DNA from small samples of blood for the diagnosis of visceral leishmaniasis. The authors have strong technical expertise in performing the RPA and qPCR assays and consequently the study was technically sound. Overall, the least expensive and simplest extraction procedure; Boil & Spin performed as well or better than the expensive and more complicated procedures; Qiagen and SwiftDx. This is a relevant observation when considering that these assays are typically performed in less than optimum conditions in LMIC countries. Overall this study provides important information in support of the KEP for the development of a field-feasible assay for detecting L. donovani in the blood.
Response: Thanks to the reviewer for positive comments.
Minor points to consider during revisions.
- The enrolment of 50 VL cases appears to be high considering the effectiveness of the KEP. What was the time frame for the enrolment of 50 VL cases?
Response: The study was conducted in the period between 2017-2018. Although Bangladesh was pretty close to achieve the elimination target. However, primary and secondary VL cases were available then.
- Past history of VL, 15/50 is 30%, not 50% as indicated in Table 1. Also, 30% with a past history seems high. Were these relapse. Is this now normal in Bangladesh?
Response: Thanks for drawing attention to the data. The table has been revised. To confirm again, 30% of the cases had past history of VL.
As per clinical definition, if someone develops VL associated symptoms within 12 months of treatment (but 6 months after treatment) then the individual should be considered as a relapse case. At present in Bangladesh, the number of VL relapse is high, whereas the primary VL cases are going down day by day. According to the national kala-azar elimination program, around 30% of the VL cases are relapse VL who got treatment in last couple of years at different treatment centers in Bangladesh.
- What was the target DNA for RPA amplification?
Response: kDNA was the target for RPA assay. Please see reference 20.
- Regarding Yield where comparisons were made by comparing ng/ul. However, the extraction volumes were different for each approach. Was this difference taken into account when calculating the yields?
Response: In this study three different extraction methods have been compared. For each of the method the blood sample volume was different and the elution volume was different as well. The concentration of the DNA was measured in nanodrop in ng/µL. For comparing three different DNA extraction methods the DNA concentration in ng/µL was taken into account. Since we did not compare the total DNA yields, therefore we did not consider the initial blood sample volume that was used for particular extraction methods. However while determining the parasite loads in different extraction methods we considered the initial blood sample volume for calculating the amount of parasite per mL of blood.
- Figure 4B indicates “Speed Xtract” Is this the same as SwiftDx (SX). These terms should be consistent throughout.
Response: The figures have been revised and “SX” term has been used to denote the SwiftDx extraction methods in all the graphs to maintain the consistency.
- Line 370; should read “47(94%) patients were positive for L. donovani DNA using qPCR and/or RPA assays
Response: The indicated line has been rephrased according to the reviewer’s suggestion in the revised manuscript.
- In the discussion, it would be helpful to elaborate how the detection of parasite in the blood will support the KEP. Will the test only be performed on positive diagnosed cases (rK39 +) or can it also be done on potential asymptomatic or family members of VL cases.
Response: Thanks for your suggestion. Indeed, the test can be applied as a surveillance tool at the post elimination period for gauging the residual infection in the endemic hotspots. For surveillance, VL cases and their household members might be under investigation to track the ongoing transmission. However, the approach surveillance should be adopted by the country specific VL elimination programs. Please see the revised section (Line 450-454).
Reviewer 3 Report
Comments and Suggestions for Authors
I acknowledge the considerable effort made by authors to compare the different molecular methods for VL diagnosis. However, in my view, there are many missing issues: A precise justification of why they chose the target of their assays, presenting in detail the primers used in the tests.
The number of parasites detected in the blood is so high that it is close to microscopic examination, and the assay fails with low parasite loads. This is a central point, since considering that the eradication campaign has been successful, most likely you are missing persistent parasites in clinically cured patients and asymptomatic carriers. Thus, the region is sitting on a bombshell, that will explode if eradication measures are relaxed.
Why not use more sensitive targets as you mention?
Why negative controls are selected from the same endemic region?
The viscosity issue of RPA is definitively a handicap for the application of the technique
Additionally, pay attention to correctly write the taxonomic names
and consistent separation of units from the ciphers like degrees, microliters etc
Author Response
Reviewer 3:
- I acknowledge the considerable effort made by authors to compare the different molecular methods for VL diagnosis. However, in my view, there are many missing issues: A precise justification of why they chose the target of their assays, presenting in detail the primers used in the tests.
Response: The leishmania species contain around 10,000-20,000 copy numbers of kDNA. Therefore, it has been the most popular target for molecular diagnosis of leishmania species. The details of primers are depicted in our previous manuscript. Please see reference 20.
- The number of parasites detected in the blood is so high that it is close to microscopic examination, and the assay fails with low parasite loads. This is a central point, since considering that the eradication campaign has been successful, most likely you are missing persistent parasites in clinically cured patients and asymptomatic carriers. Thus, the region is sitting on a bombshell, that will explode if eradication measures are relaxed.
Response: The reviewer is correct in the context of detecting the reservoirs with very low parasite burden towards successful elimination of VL. The RPA assay has an analytical sensitivity of 0.01 parasite DNA/per reaction. It seems the assay is highly sensitive for early detection of the cases. Moreover, the quantitative feature of the method leverage the monitoring of the VL/PKDL patients after treatment. To date, very few molecular methods have been developed that could achieve this analytical sensitivity. Moreover, samples with very low parasite DNA are vulnerable to freeze-thaw steps while working on downstream procedures.
In line with the reviewer’s comment, there is high likely of imminent outbreaks when the elimination efforts will be relaxed. In the elimination context, still it is necessary to treat the PKDL cases to discontinue the transmission of the parasites form these cryptic reservoirs. Several studies showed the applicability of the RPA assay in detecting and treatment monitoring of such cases. To date there has been no case definition for asymptomatic carriers. Moreover, xeno-diagnosis study could not prove that asymptomatic carriers can transmit the disease. The elimination program needs an ultra-sensitive diagnostic tool for detecting the infection early which should offer field feasibility as well. To the best of our knowledge, this RPA assay coupled with rapid DNA extraction method could capacitate the remote laboratories the most towards diagnosis of VL cases. Ultimately the elimination programs will be benefited with this low-cost point-of-need diagnostic tool to carry out efforts with limited resources at post-elimination period.
- Why not use more sensitive targets as you mention?
Response: The RPA assay is single-plex which detects one DNA target in a reaction. Most of the molecular assays being developed for leishmania are kDNA based. Therefore, our RPA assay has been developed targeting the kDNA sequence.
- Why negative controls are selected from the same endemic region?
Response: Negative controls were used to detect the clinical specificity of the assay. Endemic normal subjects were included to maintain the homogeneity among the study subjects. Moreover, it gives information regarding the hidden infection status of the apparently healthy subjects who are living the endemic communities. On the other hand, controls from non-endemic areas would not give such information, rather we would get over-estimated specificity of the assay.
- The viscosity issue of RPA is definitively a handicap for the application of the technique
Response: In our study qPCR showed only 2% extra sensitivity than that of RPA assay. The difference is negligible in terms of application of the assay, whereas the assay overcomes many challenges incurred by qPCR. Eventually the assay has promises to be implemented in large scale. The methodological challenges of the assay would be taken into account prior further application.
- Additionally, pay attention to correctly write the taxonomic names and consistent separation of units from the ciphers like degrees, microliters etc
Response: The manuscript has been revised according to the reviewer’s proposition.
Round 2
Reviewer 1 Report
Comments and Suggestions for Authors
The authors did remove the ROC graphs but they did not remove the ROC analysis entirely. In this case where you compare different combinations of extraction methods and molecular tests, a ROC analysis is just not appropriate. It could be replaced by the Youden index. Other than that the article is OK.
Comments on the Quality of English LanguageMinor corrections required
Author Response
Reviewer 1
The authors did remove the ROC graphs but they did not remove the ROC analysis entirely. In this case where you compare different combinations of extraction methods and molecular tests, a ROC analysis is just not appropriate. It could be replaced by the Youden index. Other than that the article is OK.
Response: Thank you for your comments. Considering the facts, we have excluded the ROC analysis from our manuscript.
Reviewer 3 Report
Comments and Suggestions for Authors
dear authors, you made a big effort to improve the work and deleted unnecessary information. The designation of the taxonomic names are still confusing, you must italicize when species are mentioned, but not when you just refer to Leishmania.
Yes, you mention now that k-DNA is used as a target, nevertheless, that should be an important piece of information when describing your method, which primer you used etc.
Third I am not convinced about your reasoning for choosing the controls from the same endemic area, since you claimed you would possibly detect background or low manifestations of the disease, and enhance your results
Precisely that´s what we are trying to avoid, the potential noise caused by local endemic area controls.
In your figure, from a first look, it appears that you mix the control and the infected people.
Author Response
Comment: Dear authors, you made a big effort to improve the work and deleted unnecessary information. The designation of the taxonomic names are still confusing, you must italicize when species are mentioned, but not when you just refer to Leishmania.
Response: The manuscript has been revised following the reviewer’s suggestion.
Comment: Yes, you mention now that k-DNA is used as a target, nevertheless, that should be an important piece of information when describing your method, which primer you used etc.
Response: The primer-probe used in the current study are illustrated in detail in our previously published manuscript. The citation has been added to this manuscript for the reader.
Comment: Third I am not convinced about your reasoning for choosing the controls from the same endemic area, since you claimed you would possibly detect background or low manifestations of the disease, and enhance your results
Precisely that´s what we are trying to avoid, the potential noise caused by local endemic area controls.
Response: We would like to emphasize on our justification towards inclusion of endemic controls for determining the specificity of the index diagnostic methods. In general, in any diagnostic evaluation study, cases and controls are added to determine the sensitivity and specificity respectively. Visceral Leishmaniasis (VL) is an parasitic disease which takes longer period of time to be patent with particular symptoms in infected patients. It is not unlikely that individual in the endemic areas are expose to the parasite where a very small subset of the exposed individuals develop the disease. The overarching goal of the study not only to diagnose the patients only with specific VL symptoms but also establish a point of need diagnostic tool which would be useful for surveillance of the disease even at the post elimination period of the VL. Eventually, we need a highly specific methods to clearly distinguish the true positive and false positive cases. In the current study we included the endemic controls which are already negative by serological methods (rk-39 and DAT). Therefore, the absolute specificity of the methods nullify any chances of getting false positive or background noise. In our previous studies we followed the same approach to determine the sensitivity and specificity of different diagnostic methods (PMID: 37862287; PMID: 37862287; PMID: 37862287).
Comment: In your figure, from a first look, it appears that you mix the control and the infected people.
Response: The figure-01 has been clearly defined in the study design section of the method materials. We hope readers can receive the proper information that the authors tried to convey.